# Single-Cell RNA-Seq Analysis Reveals the Acquisition of Cancer Stem Cell Traits and Increase of Cell–Cell Signaling during EMT Progression

**DOI:** 10.3390/cancers13225726

**Published:** 2021-11-16

**Authors:** Federico Bocci, Peijie Zhou, Qing Nie

**Affiliations:** 1Department of Mathematics, University of California, Irvine, CA 92697, USA; fbocci@uci.edu (F.B.); peijiez1@uci.edu (P.Z.); 2NSF-Simons Center for Multiscale Cell Fate Research, University of California, Irvine, CA 92697, USA

**Keywords:** EMT, cancer stem cells, cell–cell signaling, scRNA-seq, epithelial–mesenchymal plasticity, hybrid E/M, notch, WNT, TGFB

## Abstract

**Simple Summary:**

Phenotypic plasticity is emerging as a crucial feature across multiple axes of cancer progression. Cells undergoing the epithelial–mesenchymal transition (EMT) can fall into intermediate or hybrid epithelial/mesenchymal (E/M) cell states. Moreover, cancer cells across the EMT spectrum can exhibit the traits of cancer stem cells (CSCs) and communicate through several cell-cell signaling pathways. By integrating multiple analytical tools into a single computational framework, we investigated several single-cell RNA-sequencing (scRNA-seq) datasets and identified the emerging relationship between EMT, acquisition of CSC traits, and cell–cell communication. Our integrated analysis shows that the increase of EM plasticity correlates with high expression of CSC markers as well as intensification of cell–cell signaling between cancer cells. Furthermore, these observations are consistent across different cancer types and anatomical locations. Overall, our results shine light onto the interconnected and multi-dimensional landscape of cancer progression.

**Abstract:**

Intermediate cell states (ICSs) during the epithelial–mesenchymal transition (EMT) are emerging as a driving force of cancer invasion and metastasis. ICSs typically exhibit hybrid epithelial/mesenchymal characteristics as well as cancer stem cell (CSC) traits including proliferation and drug resistance. Here, we analyze several single-cell RNA-seq (scRNA-seq) datasets to investigate the relation between several axes of cancer progression including EMT, CSC traits, and cell–cell signaling. To accomplish this task, we integrate computational methods for clustering and trajectory inference with analysis of EMT gene signatures, CSC markers, and cell–cell signaling pathways, and highlight conserved and specific processes across the datasets. Our analysis reveals that “standard” measures of pluripotency often used in developmental contexts do not necessarily correlate with EMT progression and expression of CSC-related markers. Conversely, an EMT circuit energy that quantifies the co-expression of epithelial and mesenchymal genes consistently increases along EMT trajectories across different cancer types and anatomical locations. Moreover, despite the high context specificity of signal transduction across different cell types, cells undergoing EMT always increased their potential to send and receive signals from other cells.

## 1. Introduction

The epithelial–mesenchymal transition (EMT) stands out as one of the main molecular processes that drive cancer progression by facilitating cell migration, metabolic reprogramming, and interactions between the tumor and immune system [1,2]. EMT has been traditionally regarded as a binary transition between an epithelial state with strong cell–cell adhesion and apicobasal polarity and a mesenchymal state with motile and invasive traits. Experimental and theoretical evidence over the last decade, however, painted a more complex picture with a spectrum of intermediate, or hybrid, epithelial/mesenchymal (E/M) states [3,4]. These intermediate states play a crucial role in cancer progression and metastasis by promoting collective cell invasion, resistance to therapies, and cell proliferation [5,6].

Hybrid E/M phenotypes have been defined based on the co-expression of epithelial and mesenchymal markers, cell morphology, and propensity to collective migration [7]. Yet, we still lack a general consensus on how to define these cell states. Single-cell transcriptomics now offers unprecedented resolution on gene expression regulation during EMT, proving that many EMT-associated molecular programs are context specific, thus making a general definition even more challenging [8]. Recently, intermediate EMT states and the transitions between them have been identified from single-cell RNA-sequencing (scRNA-seq) data in various human and mouse cancers [9,10]. Many questions remain, however, on how to characterize these intermediate states and their connection with other axes of cancer progression, such as tumor-initiating ability and cell–cell signaling. To tackle these questions, here we integrate existing tools for clustering and trajectory inference from scRNA-seq data with analysis of epithelial and mesenchymal gene signatures [11]. Moreover, we evaluate cell plasticity along the epithelial–mesenchymal spectrum (or E-M plasticity) by computing an EMT circuit energy that is maximized when both epithelial and mesenchymal genes are highly expressed. Together, these methodologies provide unprecedented information on EMT progression and its intermediate states.

The importance of intermediate EMT states is underscored by their increased metastatic potential [12,13]. In a clinical setting, cancer stem cell (CSC) traits are typically defined in terms of tumor-initiating ability (such as number of metastases in vivo or tumor organoids in vitro) and resistance to therapies. Furthermore, a handful of cancer-specific CSC markers typically correlate with cancer aggressiveness and resistance, and their expression is often used as an operative definition of stemness [14]. In developmental contexts, it has been recently shown that transcriptional diversity, which is simply expressed as the number of genes expressed in a cell, decreases along developmental trajectories [15]. Similarly, a developmental single cell energy based on correlations between highly expressed genes has been recently defined and shown to decrease from pluripotent cells to lineages of differentiated cells [16]. The challenge to relate empirical (organoid formation, CSC markers) and “systems-based” (transcriptional diversity, developmental energy) definitions of stemness remains open, and holds promise to unlock better therapeutic strategies to identify highly aggressive subsets of cancer cells in vivo.

Furthermore, EMT progression can be induced by signaling between cancer cells. Juxtacrine signaling acting through physical contact between neighboring cells, such as Notch, and paracrine signaling acting through diffusible ligands, such as WNT and TGF-beta, are often altered and/or overexpressed in cancer tissues, and are linked to oncogenic potential and metastasis [17,18,19,20]. Recently, several mathematical methods have been developed to infer cell–cell communication from scRNA-seq data [21]. These methods, which typically include curated databases of ligands, receptors, and mediators of several signaling pathways, allow an investigation of the propensity to send and receive signals systematically, thus providing a complete picture of cell–cell communication between different cell states [21]. While several cell–cell communication pathways have been previously related to EMT progression on a case-by-case basis, the dynamics of cell–cell signaling during EMT has not been studied in a systematic manner yet.

Here, we develop an integrated computational framework that combines existing tools to analyze scRNA-seq and shed light onto the connection between EMT, acquisition of CSC traits, and cell–cell signaling, thus investigating several aspects of the multi-dimensional landscape of cancer progression. For our analysis, we considered datasets that are representative of several different experimental systems (normal vs. tumor), models (human/mouse, in vivo/in vitro), anatomical regions (primary tumors/CTCs/metastases), and experimental conditions for in vitro datasets (time course, spatial culture). First, we show that “system-level” measures of stemness, such as transcriptional diversity and developmental potential, do not necessarily exhibit a consistent trend along the reconstructed EMT trajectories. Conversely, the EMT circuit energy consistently increases along all identified trajectories, and often correlates well with the expression of CSC markers, such as CD24 and CD44. Moreover, we use CellChat [22] to study cell–cell signaling and discover that cells increase their propensity to send and receive signals as they progress through EMT, thus suggesting a critical role for cell–cell communication in the spatial and temporal propagation of aggressive E/M cell phenotypes. Overall, our results lay the foundation for a more quantitative identification of intermediate EMT states, and quantitatively substantiate the connection between EMT, CSC traits, and cell–cell signaling.

## 2. Materials and Methods

### 2.1. Preprocessing of Single-Cell Datasets

Datasets were preprocessed with QuanTC to select a number of top expressed genes (*n* = 3000) and discard cells that express less than a minimal percentage of genes (5%). QuanTC preprocessing selected *n* = 361 cells for the SCC dataset, *n* = 665, 1772, 2794 cells for OVCA420 under TGF-beta, EGF, and TNF induction, respectively, *n* = 2034 and 2239 cells in the inner and outer MCF10A datasets, respectively, and *n* = 2215 cells in the HNSCC dataset. For all datasets, a pseudo count was added before log-transforming the count matrix. 

### 2.2. Clustering and Trajectory Inference with QuanTC

QuanTC was used to identify cell clusters and the transition trajectories between them [9]. To select the number of clusters, QuanTC computes a cell-cell similarity matrix using the SC3 package [23]. The optimal number of clusters is established based on the maximal gap between consecutive eigenvalues. Since this study especially focuses on intermediate EMT states, we followed previous applications of the method and constraint to a minimum of three clusters to identify at least one potential intermediate state [10]. The starting cluster for trajectory inference was identified based on epithelial and mesenchymal scores of the clusters. The cluster with the highest epithelial score and lowest mesenchymal score was always chosen as the starting point of the EMT trajectory (epithelial and mesenchymal scores are defined in the following section). In the HNSCC dataset, no clear-cut epithelial state was identified; in this case, different clusters were tested as potential starting points. All other parameters for trajectory inference were kept as their original values (https://github.com/yutongo/QuanTC/blob/master/Example/QuanTC_SCC.pdf, accessed on 4 October 2021). Further details can be found in the original publication [9].

### 2.3. E-M Scoring with AUCell

Gene signatures for 108 downregulated (epithelial score) and 193 upregulated (mesenchymal score) genes were taken from a previously established gene signature derived in TGF-beta driven EMT, which has then been applied to multiple different datasets/cancer types [11] (Appendix A). These same epithelial gene signatures (downregulated EMT genes) and mesenchymal signatures (upregulated EMT genes) were applied to all considered datasets. The strength of the E and M gene signatures in each cell were evaluated using AUCell (https://bioconductor.org/packages/release/bioc/html/AUCell.html, accessed on 4 October 2021) [24]. The only free parameter in the AUCell signature pipeline is the number of genes to include to evaluate the strength of the gene signature. Due to the different sequencing depths in various datasets, this number was picked on a case-by-case basis using the AUCell_buildRankings() function (*n* = 7000 for SCC dataset, *n* = 5000 for HNSCC, all genes for OVCA420 and MCF10A datasets).

### 2.4. Calculation of Transcriptional Diversity and Developmental Potential

The transcriptional diversity [15] was previously defined as the number of expressed genes, and it is therefore computed simply by counting the number of genes with at least one mRNA count in each cell. Therefore, this quantity does not depend on how/if the count matrix is normalized. The single cell developmental energy (or developmental potential in brief) was computed with scEpath and was defined in the original publication [16]. Briefly, to compute the developmental potential, scEpath first reconstructs a gene–gene interaction network based on correlations between genes across the whole dataset. To construct this network, the default threshold was kept (tau = 0.4) to filter out connections between weakly correlated genes. In the original study, it was shown that the results only weakly depend on this parameter’s choice [16]. Then, an energy is computed based on nearest-neighbors’ correlations. To compute the scEpath energy, the steps highlighted in the original method were followed without modifications (https://github.com/sqjin/scEpath, accessed on 4 October 2021). Therefore, both the transcriptional diversity and scEpath developmental potential consider all genes detected in any given dataset.

### 2.5. Quantification of EMT Circuit Energy

To quantify an EMT circuit energy, we restricted the above-mentioned scEpath algorithm to the selected list of 108 epithelial genes and 193 mesenchymal genes previously used to compute the epithelial and mesenchymal scores with AUCell (see Section 2.3). Specifically, the datasets were filtered to select only the downregulated or upregulated EMT genes [11] (i.e., the same genes introduced in Section 3 to compute the E-M scores). Subsequently, scEpath was applied to this restricted EMT count matrix, thus creating a circuit of interacting EMT-related genes. The full gene list can be found in the Appendix A. Since the scEpath energy is maximized when most genes in the inferred network are highly expressed, the EMT circuit energy is expected to be maximized in cell states with high E-M plasticity (i.e., when both E and M genes are expressed).

### 2.6. Analysis of Cell–Cell Communication

Cellchat was used to study cell–cell communication between cell types [22]. CellChat quantifies the propensity to behave as a sender or receiver for numerous cell–cell signaling pathways included in the KEGG dataset. For datasets where different cases could be compared (TGF-beta vs. EGF vs. TNF induction in OVCA420, inner vs. outer in MCF10A, tumor vs. lymph nodes in HNSCC), CellChat analysis was first performed in each case separately, and then the mergeCellChat() function was used to compare them. Raw count matrices were provided as input and normalized with the built-in normalization function. For this analysis, all genes in the dataset were considered to facilitate identification of multiple pathways. All parameters were kept as their original values as found in the package walkthroughs (https://github.com/sqjin/CellChat, accessed on 4 October 2021). Further information on the package can be found in the original publication [22].

### 2.7. Data Availability

All datasets are available in the NCBI Gene Expression Omnibus. The squamous cell carcinoma (SCC) dataset from Pastushenko and collaborators [25] is available under accession number GSE110357. The time course of OVCA420 data under various inducers from Cook and collaborators [8] is available under accession number GSE147405. MCF10A data from McFaline-Figueroa and collaborators is available under accession number GSE114687 [26]. Head and neck squamous cell carcinoma (HNSCC) data from Puram and collaborators is available under accession number GSE103322 [27].

## 3. Results

### 3.1. Plasticity of Intermediate EMT States from In Vivo Squamous Cell Carcinoma CTCs 

To investigate intermediate EMT states and their connection with cancer stem cell (CSC) traits and cell–cell signaling, we first considered scRNA-seq of 361 circulating tumor cells (CTCs) from a squamous cell carcinoma (SCC) mouse model [25]. We have previously shown that unsupervised clustering with QuanTC leads to the identification of four cell types in the SCC dataset [10], which can be visualized in a low-dimensional projection space (Figure 1A and Appendix A). To quantify the location of the different clusters along the EMT spectrum, we used AUCell [24] to evaluate a comprehensive gene signature for genes that are downregulated (epithelial signature) or upregulated (mesenchymal signature) during TGFB-driven EMT [11] (Appendix A). Compared to the direct comparison of gene expression levels, AUCell offers a more robust estimation of gene signature strength that is based on the area under the curve considering the gene expression distribution in an entire dataset 24. This approach has been previously applied to bulk RNA-seq datasets to successfully identify hybrid epithelial/mesenchymal cell phenotypes in breast cancer [28]. One of the four clusters exhibits the largest epithelial signature and lowest mesenchymal signature and is thus identified as the epithelial (E) cluster (*n* = 86 cells). Conversely, the other three clusters all exhibit higher mesenchymal scores and share a mutual overlap in the E-M score space (Figure 1B). To gain more insight into the biological role of these three states, we reconstructed the pseudotime ordering and EMT trajectories starting from the epithelial state with QuanTC. Consistently, the dominant path involving the highest number of cells starts from the epithelial clusters, passes through two intermediate states, and finally reaches the state with the lowest epithelial signature (Figure 1B). Other less frequent trajectories are possible, including trajectories that pass through only one of the two intermediate states, or go directly from the E cluster to the terminal cluster (see Appendix A for details). Importantly, all these trajectories share the same terminal cluster, which is thus identified as the mesenchymal state (M, *n* = 80 cells), while the two remaining clusters are considered as intermediates I1 (*n* = 130 cells) and I2 (*n* = 65 cells) (Figure 1B). Consistently, analysis of the highly expressed genes along the predicted trajectory highlights typical epithelial genes, such as Epcam and Krt14 at the beginning of the transition (Appendix A). To check the robustness of the clustering and EMT scoring, we further test their dependence on (i) the minimal fraction of expressed genes per cell required during preprocessing and (ii) the number of top expressed genes selected, showing that the results remain largely unaltered upon local variation of these parameters (Appendix A).

Next, we tested the pluripotency of SCC cells in the four clusters by computing (1) the transcriptional diversity and (2) the developmental energy using scEpath (see Section 2.4). Interestingly, both quantities decrease from the E to I1, to then increase again and reach a maximal value in the M states (Figure 1C). Conversely, the EMT circuit energy monotonically increases along the trajectory (Figure 1C). Therefore, while the total transcriptional activity quantified by either diversity or developmental energy temporarily decreases along the E-I1-I2-M transition path, the plasticity of the EMT regulatory network per se continues to increase and reaches its maximum in the terminal M state. Intuitively, this can be understood by observing that the transition trajectory is characterized by a significant increase of the mesenchymal signature but only a small decrease in the epithelial signature, potentially suggesting that the terminal state that we previously labeled as mesenchymal (M state) has hybrid E/M properties instead. This hypothesis is strengthened when observing that only ‘quasi-mesenchymal’ cells with hybrid morphology but not fully mesenchymal cells were reported in the original study [25].

To further validate our clustering results, we analyzed the expression of aggressiveness markers. In the original study, cell subpopulations were identified based on the expression of three markers: CD51, CD61, and CD106. Therefore, we further analyzed the composition of the E, I1, I2, and M clusters based on marker expression by clustering the cells based on high or low expression of CD51, CD61, and CD106 independently (Figure 1D). Consistent with the original study, the fraction of triple negative (TN) cells decreases with increased EMT stage, while the fraction of cells expressing one or more markers progressively increases along the EMT trajectory. In particular, the fraction of cells that expresses all three markers (triple positive, TP) is maximized in the M cluster. 

Finally, we used CellChat to characterize the contribution of the different cell types to cell–cell signaling. This analysis highlights the activation of several signaling pathways, including many “usual suspects” often implicated in cancer progression, such as TGF-beta, Notch, and WNT (Appendix A). The propensity to behave as a sender or receiver of signaling for cells in different clusters can be summarized in a low-dimensional projection (Figure 1E). Interestingly, E cells mostly behave as a signal receiver, while intermediate I1 cells mostly behave as senders. Conversely, intermediate I2 and M cells that are more advanced in the EMT trajectory behave both as strong senders and receivers (Figure 1E). When inspecting specific pathways, we find that E cells are fairly inactive in terms of TGF-beta signaling, which is instead strong on the I2 and M clusters (Figure 1F). Interestingly, the intermediate state I2 exhibits the strongest Notch and WNT signaling, while the terminal M state is both a weaker sender and receiver. (Figure 1G,H). Interestingly, in the original experiment, it was observed that early EMT transition states were as capable of generating metastases as the more mesenchymal states [25]. Therefore, it is not surprising that signaling pathways, such as Notch and WNT, which are strongly implicated in cancer aggressiveness, are highly expressed in intermediate states. When breaking down the contribution of specific ligand-receptor pairs, we discover that JAG ligands provide the strongest contribution to Notch signaling, while DLL ligands are weakly expressed (Appendix A). Consistently, JAG ligands have been previously correlated with lower overall patient survival and higher metastatic potential [29,30]. Interestingly, while E cells generally have a low ‘outgoing interaction score’, they behave as strong senders of Notch and WNT signaling, thus suggesting strong signaling between cells in different EMT states.

Overall, the integrated analysis of SCC CTCs suggests that cell states with maximal EMT circuit energy also tend to maximize “systems-level” measures of stemness, such as cell developmental energy and transcriptional diversity, as well as facilitating crosstalk between cancer cells by acting both as senders and receivers of several cell–cell communication pathways. However, many signaling pathways implicated in cancer metastasis are maximized in the intermediate EMT states. To test whether some of these findings are conserved, we next turn to the analysis of other datasets from different cancer types and experimental conditions.

### 3.2. EMT Time Course of an Ovarian Cancer Cell Line Highlights Transient Activation of EMT Plasticity and Cell–Cell Signaling

To explicitly investigate the temporal evolution of EMT plasticity and cell–cell signaling, we consider a recent time course of TGF*β*-induced EMT in OVCA420 ovarian cancer cells in vitro [8]. In this experiment, cells were exposed to an EMT-inducing signal at t = 0 and scRNA-seq was performed on cells at subsequent time points for 7 days of EMT induction (0 days, 8 h, 1 day, 3 days, 7 days) and for 3 days after EMT-signal removal (8 h, 1 day, 3 days).

First, QuanTC identifies four states connected by a trajectory that starts from the most epithelial state (higher E score, lower M score, *n* = 140 cells) and ends at the most mesenchymal state (lower E score, higher M score, *n* = 87 cells) (Figure 2A,B and Appendix A). Similar to the previous case, the intermediate states of the trajectory are labeled I1 and I2 (*n* = 138 cells and *n* = 166 cells, respectively). Differently from the SCC CTCs in vivo, however, neither the transcriptional diversity nor the scEpath developmental energy show a consistent trend along the trajectory, and thus cannot be considered as a good indicator of pluripotency to distinguish between cell types (Figure 2C). While both quantities are higher in the terminal state (M) as compared to the starting state (E), they are both maximized in intermediate states along the trajectory (Figure 2C). Conversely, the energy of the EMT circuit increases consistently along the E-I1-I2-M path (Figure 2C). It is worth pointing out that, although we used the same labels to identify clusters across different datasets (i.e., E, I1, I2, M), it is not straightforward to establish a direct comparison. In other words, the clusters labeled as E or M in different datasets might have different properties, which can be distinguished by analyzing the EMT signature and cell–cell communication.

Next, to validate our clustering results, we checked how our identified cell states are distributed against the physical time of the time course. Cells sequenced at the moment of EMT induction (t = 0 days) are mostly epithelial, while the proportions of cells in the intermediate states (I1 and I2) progressively increase at later time points. Finally, most cells are mesenchymal (M) after one full week of EMT induction (Figure 2D). Interestingly, the cell population after 3 days from TGF-beta removal does not reverse completely to the original cell state proportions but is rather a mixture of the four phenotypes (Figure 2D). Different mechanisms can explain this finding. The simpler explanation is that complete MET cannot be achieved within 3 days from TGF-beta removal. Alternatively, the more highly populated intermediate states can be explained by hysteresis in the EMT transitions, which has been previously reported in TGF-beta-driven lung metastatic colonization [31]. In other words, the intermediate and mesenchymal states can be stable even in the absence of EMT inducers, and therefore remain populated even once the TGF-beta signal is removed. Longer remission time courses would be necessary in the future to distinguish between the two mechanisms.

Consistent with the cell fraction analysis, the EMT circuit energy increases from t = 0 days, maintains a maximal value between t = 3 days and t = 1 day of remission, and then decreases to a lower value after 3 days of remission. Therefore, these results support a model where the initially epithelial cell population shifts toward intermediate and mesenchymal cell states due to TGF-beta and then undergoes MET to reverses back to less mesenchymal states once the EMT-inducing signal is removed.

Remarkably, the expression of standard ovarian cancer CSC markers, such as CD24 and CD44, gradually increases along the EMT trajectory. Accordingly, in physical time, they reach a maximal expression after around 7 days of EMT induction, before decreasing to lower levels once the TGF-beta signal is removed (Appendix A). 

Moreover, cells at the beginning (t = 0, 8 h, 1 day) and end (t = 3 days after remission) of the time course behave as weak senders and receivers of cell–cell signaling. Conversely, cells between t = 3 days and t = 1 day of remission behave both as a strong sender and receiver, thus suggesting a strong relationship between EMT and cell–cell signaling (Figure 2F). Therefore, it can be concluded that TGF*β*-induced EMT transiently activates CSC markers’ expression and several cell–cell signaling mechanisms. While WNT was not found to be significantly expressed in the CellChat analysis, both TGF-beta and Notch were weak at the beginning and end of the time course, while peaking between t = 7 days and t = 1 day of remission Figure 2G and Appendix A). Specifically, we found again that Notch signaling is carried primarily by Jagged ligands (JAG1, JAG2) despite the different organisms (human vs. mouse) and conditions (in vitro vs. in vivo) (Appendix A).

Overall, induction of EMT through TGF-beta transiently increased OVCA420 cells EMT circuit energy, CSC markers, and cell–cell signaling, thus confirming many of the trends previously established in the SCC dataset. To further test the proposed connection between EMT activation and cell–cell signaling, we next compared EMT induction time courses of OVCA420 cells with different EMT inducers.

### 3.3. Comparison of Different EMT-Inducing Signals in OVCA420 Cells Confirms the Relationship between EMT Progression and Activation of Cell–Cell Signaling

In the original experiment, OVCA420 cells were also exposed to two additional EMT-inducing signals, TNF and EGF [8]. We have previously shown that TNF- and EGF-driven EMT leads to less consistent and synchronized EMT transitions compared to TGF*β* [10]. When applying the same analytical pipeline, we identified three clusters (E, I, and M) for both datasets (*n* = 925, *n* = 1468, *n* = 331 and *n* = 370, *n* = 847, *n* = 407 cells in E, I, and M states for TNF and EGF datasets, respectively) (Figure 3A,D and Appendix A). Differently from the TGF-beta driven EMT, the cell fractions of E, I, and M cells do not change significantly during the EGF and TNF time courses (Figure 3B,E). The difference between TGF-beta, EGF, and TNF induction of EMT can also be appreciated by examining the EMT circuit energy. In all three datasets, the EMT circuit energy significantly increases from the E to the M cell state, thus confirming the presence of different EMT phenotypes with increasing EM plasticity (Figure 3C,F). TNF and EGF, however, only lead to a modest and largely statistically insignificant increase of EMT circuit energy during the time course (Figure 3G,H). Therefore, while TGF-beta changed the stability of different EMT phenotypes by destabilizing the E state and stabilizing the M state, EGF and TNF induction are not sufficient to dramatically modify the E-M landscape. Interestingly, the overall developmental energy increases in parallel with the EMT circuit energy, whereas the transcriptional diversity does not follow a clear trend (Appendix A). Therefore, when considering OVCA420 cells under all three EMT induction conditions (TGF*β*, EGF, TNF), the EMT circuit energy is the only variable that exhibits a consistent trend along the reconstructed EMT trajectories. Moreover, while the E clusters of the EGF and TNF datasets show many common marker genes (LGALS4, LCN2, BPIFA2, GSTA1), I and M cells show virtually no overlap in terms of top marker genes, thus highlighting a high context specificity in the EMT trajectories (Appendix A).

Comparing cell–cell interactions across the three datasets further shows that EGF and TNF signals are not sufficient to enhance the sender and receiver propensities of OVCA420 cells (Figure 3I,J), thus supporting the association between EMT progression and the ability to activate cell–cell communication between cancer cells. In general, the EGF and TNF cases exhibit much weaker cell–cell signaling, and many pathways typically associated with EMT progression, such as TGFB, Notch, TFN, and IL1, were exclusively expressed in the TGFB-driven EMT (Figure 3K and Appendix A).

Overall, comparing the time courses suggests that OVCA420 cells exhibit a strong heterogeneity with multiple cell states, even when no EMT induction is applied. TGF-beta-driven EMT modifies this landscape and introduces new, more mesenchymal states, whereas EGF-driven and TNF-driven EMT only minorly change the proportions of cell states. Moreover, comparing EMT circuit energy and cell–cell communication highlights the strong connection between EMT progression and signaling. While cells become stronger senders and receivers of cell–cell signals under TGF-beta, their behavior remains largely unaltered under EGF or TNF induction.

### 3.4. Space-Dependent EMT Phenotype Fractions in MCF10A Cells

So far, we have considered datasets where, at most, only temporal information was available. EMT, however, is heavily regulated by signaling between cells and the surrounding environment. Therefore, it is expected that cells in different spatial locations respond to different stimuli coming from the physiological or tumor microenvironment. To test how spatial location modulates the association between EMT and cell–cell signaling, we next consider spontaneous EMT in MCF10A mammary epithelial cells that were plated at the center of a culture dish and were then free to migrate toward the margins of the plate [26]. In the original experiment, the “outer” cell population developed a stronger mesenchymal migratory trait, while the “inner” population remained relatively more epithelial [26].

QuanTC identifies three clusters in both the inner and outer cell populations (Figure 4A,D and Appendix A). In both cases, a first cluster has both the highest epithelial score and lowest mesenchymal score and is thus identified as the epithelial state and start of the EMT trajectory (*n* = 647 and *n* = 244 cells for inner and outer, respectively). While the E state is well separated in the (E,M) score space, the remaining two clusters share a consistent overlap. Based on trajectory reconstruction with QuanTC, we distinguish between an intermediate state (I, *n* = 1010, 1106 cells) and a terminal mesenchymal state (M, *n* = 377, 889 cells) (Figure 4B,E and Appendix A). 

Moreover, the transition from the E to I state increased the developmental potential, but not the transcriptional diversity, in both inner and outer cells. Conversely, the transition from I to M increased both the transcriptional diversity and developmental potential (Figure 4C,F). Similar to all other datasets, the EMT circuit energy increased along the trajectory, thus following the same trend of the developmental potential (Figure 4C,F). Therefore, even though not all stemness measures increased in a consistent fashion, they were all maximized in the terminal M state in both the inner and outer cell populations.

Quite interestingly, the inner and outer datasets have a similar fraction of cells belonging to the intermediate (I) state. While the inner dataset has a larger fraction of E cells, the outer dataset has a larger fraction of M cells (Figure 4G). Therefore, while both inner and outer cells can undergo spontaneous EMT and access the I and M states, the cell distribution is shifted toward the mesenchymal end of the spectrum in the outer cell population, where perhaps signaling between cells and the local microenvironment acts as an EMT inducer that stabilizes the M state and destabilizes the E state.

In terms of cell–cell signaling, we again observe a clear trend between EMT progression and communication as both incoming and outgoing interaction strengths increase along the E-I-M axis (Figure 4H and Appendix A). While this trend is clearly observed in both cell populations, outer cells have an overall lower cell–cell signaling strength across all three clusters (Figure 4I). In other words, while it is true that cell–cell signaling propensity increases along the EMT trajectory regardless of spatial location, outer cells seem to have weaker cell–cell communication capabilities. When further breaking the signaling down based on cell types, we discover that signaling involving the E and M cell types is generally weaker in the outer cells, while communication between I cells is stronger (Appendix A).

Systematically analyzing cell–cell signaling highlights a handful of pathways that are exclusively expressed in the inner cell population, including CD99, progranulin (GRN), and ephrin (EphA) (Appendix A). Interestingly, all these pathways have been previously implicated in oncogenesis [32,33,34]. Conversely, pathways highly expressed in other datasets, such as Notch, WNT, and TGF, are not detected here, perhaps due to the non-tumorigenic nature of these cells. Moreover, outer cells show a slightly weaker signaling through desmosomes, which might be explained by the higher fraction of mesenchymal cells with low cell–cell adhesion in the outer region (Appendix A).

Overall, analysis of the MCF10A inner and outer datasets highlights agreement between EMT progression, systems-level measures of stemness, and EMT circuit energy. Furthermore, comparison between inner and outer cell populations highlights the different stability of EMT phenotypes and propensity to cell–cell communication.

### 3.5. Simultaneous Acquisition of Epithelial and Mesenchymal Traits in Head and Neck SCC

Finally, we consider a head and neck squamous cell carcinoma (HNSCC) dataset with sRNA-seq data from both primary tumors and lymph node metastases [27]. QuanTC identifies three clusters (Figure 5A and Appendix A). Differently from all previous datasets, however, the epithelial and mesenchymal signatures of the clusters do not anti-correlate; rather, a first cluster has both low E and M scores (S1, *n* = 661 cells), a second cluster has high E score but low M score (S2, *n* = 267 cells), while a third cluster (S3) has both high E and M scores (Figure 5B and Appendix A). While it is typically expected that epithelial and mesenchymal signatures anticorrelate, it has been previously shown that induction of certain pathways, such as NRF2, can simultaneously increase the expression of both E-cadherin and Zeb in both monolayer culture and wound healing assays of RT4 bladder cancer cells [35,36]. 

Given the lack of a well-defined epithelial state, we tested the trajectories predicted by QuanTC when picking different clusters as a starting point. When picking the S1 cluster as a starting point, QuanTC identifies an irreversible transition from the S1 clusters to the S3 state. Moreover, S2-to-S3 and S3-to-S2 transitions are identified when picking S2 and S3 as the starting states, respectively (Figure 5B and Appendix A). Therefore, the QuanTC results suggest a direct transition from S1 to S3, where both E and M traits are enhanced simultaneously. Conversely, back-and-forth transitions between the S2 and S3 states maintain epithelial traits while gaining and losing mesenchymal traits. 

Next, we compared different measures of pluripotency and stemness. First, all three clusters exhibit a fairly similar transcriptional diversity, whereas the scEpath potential is maximized by the cluster S2 (Figure 5C). Interestingly, our calculation of EMT circuit energy provides an opposite outcome, i.e., the S2 cluster has the lowest EMT circuit plasticity while the cluster S3 has maximal EM plasticity because it maximizes both epithelial and mesenchymal gene expression (Figure 5C). Therefore, the overall pluripotency expressed by the transcriptional diversity and developmental potential do not go hand in hand with EMT circuit energy. Furthermore, no specific trend was observed when inspecting standard HNSCC CSC markers for this dataset. While CD44 expression is maximized in the S1 and S3 clusters, and thus correlates with EMT circuit plasticity, OCT4 and ALDH are minimized in cluster S3 while being lowest in S2 (Appendix A). 

Moreover, both tumor and lymph node cells are represented in all three clusters, thus showing that clustering does not solely depend on anatomical origin. Specifically, the cluster S1 is a mixture of tumor and lymph cells, cluster S2 is predominantly composed by lymph node cells while cluster S3 is predominantly composed by tumor cells (Figure 5D). Interestingly, when comparing cells based on their anatomical origin rather than predicted clustering, cells from lymph node metastases have a similar epithelial score and a slightly lower mesenchymal score compared with primary tumor cells (Appendix A). 

Finally, comparing cell–cell signaling shows that the different clusters behave similarly in tumor and lymph nodes. The S2 cluster, which has lower EMT circuit energy, is both a weak sender and receiver of cell–cell signaling, while the S3 cluster, which has the highest EMT circuit energy, has consistently high scores both as a sender and receiver in both tumor and lymph nodes (Figure 5E,F). Therefore, even though the cell states do not follow the “typical” EMT progression scenario where E and M traits anticorrelate, the relationship between EMT circuit energy and cell–cell signaling seems to emerge as a more general concept.

In general, lymph node cells show a higher propensity to cell–cell signaling. In other words, even though the correlation between EMT circuit energy and cell–cell signaling is observed in both tumor and metastases, lymph node cancer cells have consistently higher incoming and outgoing interaction strengths (Figure 5G). When focusing specifically on pathways that have been previously implicated with cell plasticity and stemness, we found a good correlation between the EMT circuit energy calculations and CellChat results. Both Notch and Wnt signaling are overall stronger in the S1 and S3 clusters and lower in the S2 cluster. Overall, Notch and Wnt are also higher in metastatic lymph cells compared to tumor cells, in good agreement with the pro-metastatic role often associated with these pathways. Strikingly, while JAG and DLL ligands almost equally contribute to Notch signaling in the tumor, JAG1 is the main contributor to Notch signaling in lymph node cells, thus confirming once more the relevance of JAG ligands in metastasis (Appendix A). Moreover, lymph cells highly express other signaling pathways that can be ultimately connected to interactions between cancer cells and T cells, such as APP, CD99, MHC-I, and CD46 (Appendix A).

Overall, our analysis of HNSCC cells identifies a strong correlation between EM plasticity and cell–cell signaling, whereas the connection between systems-level and marker-based measures of stemness is not as clear as all other considered datasets. 

## 4. Discussion

Identifying and characterizing intermediate epithelial/mesenchymal (E/M) states remains a difficult yet crucial task to advance our understanding of cancer progression [7]. Here, we devised a computational pipeline that integrates QuanTC, a method for scRNA-seq clustering, and trajectory inference [9,10], with analysis of biologically relevant gene signatures of downregulated (E score) and upregulated (M-score) EMT genes performed with AUcell [11,24]. Compared to previous iterations of the QuanTC method, here we used gene signature analysis to (i) identify the most epithelial state as the starting point of the transition and (ii) evaluate the dynamics of epithelial and mesenchymal traits along the predicted trajectories. In general, we observed a consistent trend, where trajectories pass through states with intermediate E and M gene signature scores and reach a terminal state with the lowest E score or highest M score. To further decode the “position” of these intermediate states along the EMT spectrum, we devised an EMT circuit energy that is maximized when both epithelial and mesenchymal genes are highly expressed. Remarkably, the EMT circuit energy increased along all the inferred EMT trajectory, thus potentially providing a quantitative measure to estimate EMT progression. Recently, intermediate EMT states have been identified by mapping gene expression data onto the states of an EMT Boolean network model to predict breast cancer metastatic risk [37]. Here, we took an even more general approach to show a similar emerging response of the EMT circuit across multiple cancer types. 

EMT progression is often associated with the acquisition of cancer stem cell (CSC) traits [38,39]. Recently, disseminating hybrid E/M CSCs that co-express EpCam, Vimentin, and CD24 have [37] been observed in human cancer specimens, thus proving that a hybrid E/M CSC-like state is predictive of metastasis [40]. Moreover, a recent time course of MCF10A cells under TGF-β1 induction in vitro identified subsets of intermediate hybrid epithelial/mesenchymal cells whose gene signature correlated with poor survival [41]. Similarly, metastatic cells sequenced from a pancreatic cancer mouse model are distributed along an EMT continuum, with late hybrid epithelial/mesenchymal cells exhibiting the highest metastatic potential [42]. In parallel, simple measures of cell pluripotency based on transcriptional diversity, or more advanced metrics, such as developmental potential based on the energy of an inferred gene–gene interaction network, have been successfully introduced in developmental contexts [14,16]. Given the connection between EMT progression and CSC traits, we inspected the behavior of these quantities along the reconstructed EMT trajectories. In general, we observed a good correlative trend where terminal mesenchymal states exhibited higher transcriptional diversity and developmental potential compared to their epithelial counterparts. However, these measures rarely showed a consistent trend across the trajectories. For instance, both transcriptional diversity and developmental potential first decreased and then increased along the EMT trajectory in Squamous Cell Carcinoma (SCC) circulating tumor cells (CTCs), while showing the opposite trend in OVCA420 cells under TGF-beta induction.

Solid cancers are rarely composed of similar cells with identical epigenetic traits. Rather, non-genetic determinants diversify cell populations into distinct phenotypes or cell states with various degrees of metastatic potential and resistance to various therapies [43,44]. Specifically, phenotypic heterogeneity along the EMT spectrum, or EM plasticity (EMP), is emerging as a distinctive feature of primary tumors [45], CTCs [46,47], and in the relation between tumors and CTCs [48]. This heterogeneity can arise due to diverse signaling cues that induce EMT in a spatio-temporal-dependent manner, as seen for instance in time course or wound healing experiments [36,41]. Here, we analyzed two dataset where such spatial or temporal variability could be appreciated in vitro: a time-course of EMT driven by external signals and a 2D culture where EMT is spontaneously activated at the free end of the cell colony, conceptually similar to a wound healing assay [8,26]. In both cases, we observed a shift toward more mesenchymal cell states as time progressed (i.e., after more time under EMT induction) or as cells were closer to the free end (inner vs. outer). However, a strong spatial/temporal heterogeneity was observed with the coexistence of multiple cell states along the EMT spectrum. In the time course, intermediate EMT states were observed at the beginning of the experiment and some epithelial cells were still observed after one full week of EMT induction. Similarly, in the 2D MCF10A culture, epithelial, intermediate, and mesenchymal cells co-existed, and only their relative proportions changed from the inner to outer colony. While our analysis reveals the multistability of different EMT phenotypes, we are still far from understanding the spatial organization of these cells types in vivo. Combining high-throughput analytical tools with spatial transcriptomics is a promising new direction with the potential to decode this complex spatial organization [49].

Finally, we integrated our analysis with CellChat to test the cell–cell communication propensity between cells in different EMT states [22]. Remarkably, all considered datasets showed a consistent trend where EMT progression increases the propensity to both send and receive signals from other cells. While several “usual suspects” often implicated in cancer progression, such as TGF-beta, WNT, and Notch, were significantly expressed across multiple datasets, this association emerged as a general trend independently of the specific pathway. Remarkably, the connection between EMT circuit energy and cell–cell signaling strength was conserved even when a clear EMT trajectory was not identified, such as in the case of head and neck squamous cell carcinoma (HNSCC). Cell–cell communication was previously linked to EMT progression and metastasis on a pathway-specific basis. For instance, in the context of breast cancer, the Notch ligand JAG1 correlates with lower overall survival and triple-negative breast cancer organoid formation in vitro [30,50], while EMT progression can be activated in a WNT/beta-catenin-dependent manner [51]. Here, we demonstrate this connection at a more systematic level by considering hundreds of pathways curated in the CellChat database [22]. While reconstruction of cell–cell signaling from scRNA-seq data has grown exponentially in recent years [21], we acknowledge many limitations in our current analysis. For instance, most methods for cell–cell signaling reconstruction cannot model the physical distance between cells, which might cut off communication between faraway cells. Moreover, cells undergoing EMT gradually lose their cell–cell adhesion, thus possibly weakening pathways that rely on cell–cell contact, such as Notch or desmosomes. Therefore, developing theoretical methods that better account for these limitations will in the future offer an even more precise picture of cell–cell communication during EMT. Finally, cell–cell signaling takes place at the protein level; therefore, it is tacitly assumed that mRNA counts offer an accurate picture of their corresponding protein expression. Different technologies for direct protein level measurements, such as flow cytometry, will be required in the future to offer more biological context and demonstrate the validity of the predicted EMT–signaling association on a pathway-specific and/or tumor-specific basis.

In this study, we considered several scRNA-seq datasets representative of different anatomical locations, tumor stages, and experimental conditions. The emergence of conserved traits within such a diverse pool suggests the existence of many conserved features in the interconnections between EMT, CSC traits, and cell–cell signaling. Further analysis within the context of specific tumor types will be necessary, however, to translate these exciting observations into quantitative predictors in a clinical setting.

## 5. Conclusions

Cancer metastasis remains an insurmountable clinical challenge that accounts for nearly 90% of cancer-related deaths [52]. Deciphering the relation and crosstalk between different axes of cancer progression, including EMT, acquisition of cancer stem cell traits, and cell–cell signaling, is a crucial challenge to understand cancer progression from a systems perspective. Here, we combined multiple tools into an analytical pipeline for scRNA-seq data that combined trajectory inference, analysis of biologically relevant gene signatures and CSC markers, and analysis of cell–cell communication to decode this complex multi-dimensional landscape. Despite the high context specificity of different cancer types, we highlight many emerging properties. While standard stemness measures applied in developmental biology do not necessarily capture the acquisition of CSC traits during EMT progression, the plasticity of an EMT circuit quantified by a circuit energy is a reliable measure that correlates well with standard CSC markers. Moreover, high EMT plasticity correlates with enhanced cell–cell signaling, thus suggesting that crosstalk between highly metastatic cancer cells is a general feature of cancer invasion.

## Figures and Tables

**Figure 1 cancers-13-05726-f001:**
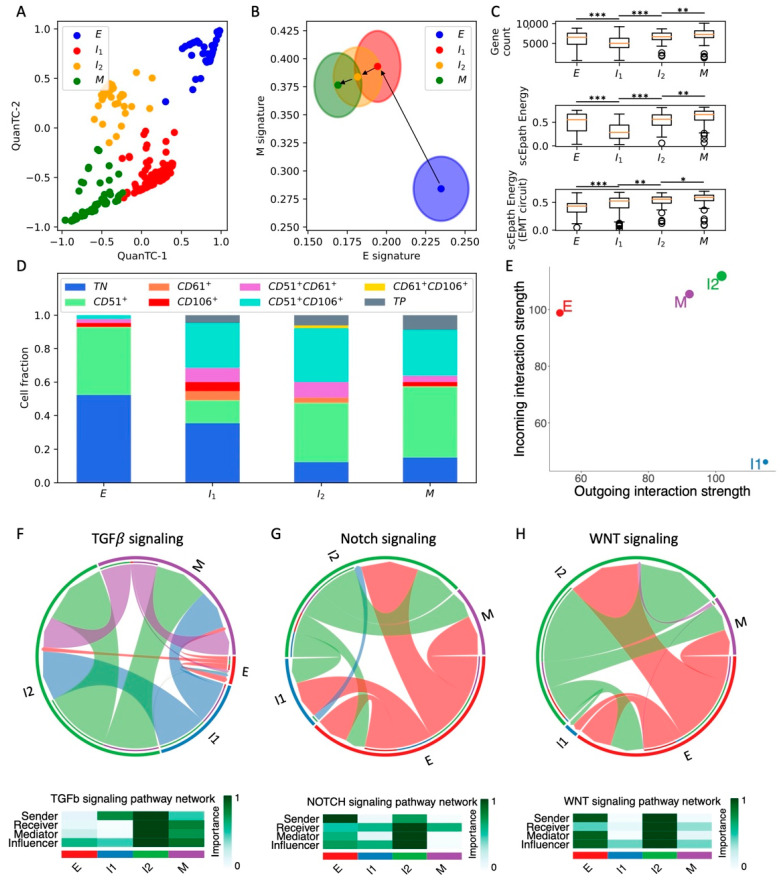
Analysis of EMT, stemness, and cell–cell signaling in the SCC dataset. (**A**) Low-dimensional representation of cells in the SCC dataset. *x*- and *y*-axis represent QuanTC coordinates. (**B**) Projection of clusters in the E-M signature space. The center and radius of each circle represent the average scores and standard deviations of the epithelial and mesenchymal signatures within each cluster. Black arrows show the dominant transition trajectory predicted by QuanTC. (**C**) Boxplots depicting the total gene count (i.e., transcriptional diversity, top), scEpath developmental potential (middle), and scEpath EMT circuit energy (bottom) in the four clusters. (**D**) Cluster composition based on CD51, CD61, and CD106 markers. (**E**) Low-dimensional projection of cell clusters in an incoming/outgoing signaling strength computed with CellChat. (**F**–**H**) Chord diagrams representing the signaling between clusters through Notch (**F**), WNT (**G**), and TGF-beta (**H**) reconstructed with CellChat. Heatmaps quantify the role of each cluster as a sender, receiver, mediator, and influencer. *, **, and *** indicate a *p*-value < 0.1, 0.01, and 0.001 on a one-tailed *t*-test, respectively.

**Figure 2 cancers-13-05726-f002:**
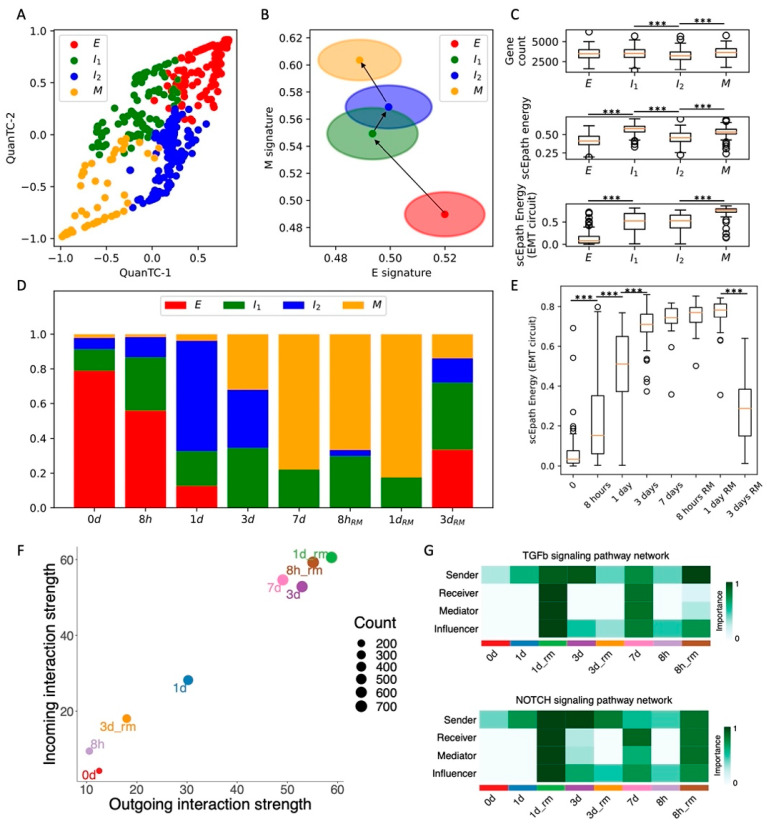
Analysis of the OVCA420 dataset under TGF-beta-driven EMT induction. (**A**) Low-dimensional representation of cells in the OVCA420 dataset under TGF-beta induction. (**B**) Projection of clusters in the E-M signature space. Black arrows show the dominant transition trajectory predicted by QuanTC. (**C**) Boxplots depicting transcriptional diversity (top), developmental potential (middle), and EMT circuit energy (bottom) in the four clusters. (**D**) Fraction of E, I1, I2, and M cells at different time points during the time course. (**E**) Boxplot of EMT circuit energy during the time course. (**F**) Low-dimensional projection of cell clusters in an incoming/outgoing signaling strength computed with CellChat. (**G**) Heatmaps quantify the role of each cluster as a sender, receiver, mediator, and influencer of TGF-beta signaling (top) and Notch signaling (bottom). *** indicates *p*-value < 0.001 on a one-tailed *t*-test.

**Figure 3 cancers-13-05726-f003:**
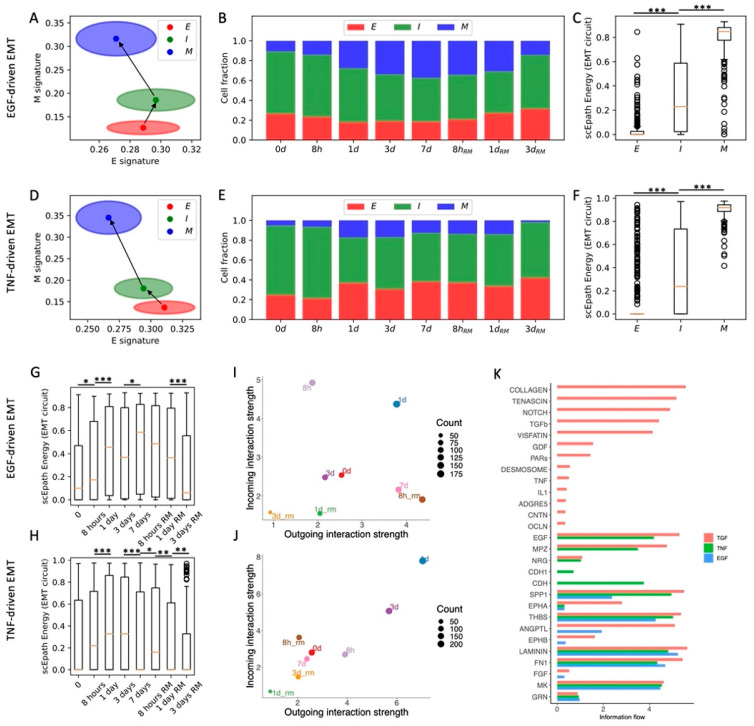
Comparison between TGF-beta-, EGF-, and TNF-driven EMT in OVCA420 cells. (**A**) Projection of clusters of OVCA420 cells under EGF-driven EMT induction in the E-M signature space. Black arrows show the dominant transition trajectory predicted by QuanTC. (**B**) Fraction of E, I, and M cells at different time points during the time course. (**C**) Boxplots depicting EMT circuit energy in the three clusters. (**D**–**F**) Same as (**A**–**C**) for OVCA420 cells under TNF-driven EMT. (**G**) Boxplot of EMT circuit energy during the time course of EGF-driven EMT. (**H**) Same as (**G**) for TNF-driven EMT. (**I**) Low-dimensional projection of cell clusters in an incoming/outgoing signaling space for EGF-driven EMT. (**J**) Same as (**I**) for TNF-driven EMT. (**K**) Comparison of signaling pathway information flow between TGF-beta (red), EGF (green), and TNF-driven (blue) EMT. *, **, and *** indicate a *p*-value < 0.1, 0.01, and 0.001 on a one-tailed *t*-test, respectively.

**Figure 4 cancers-13-05726-f004:**
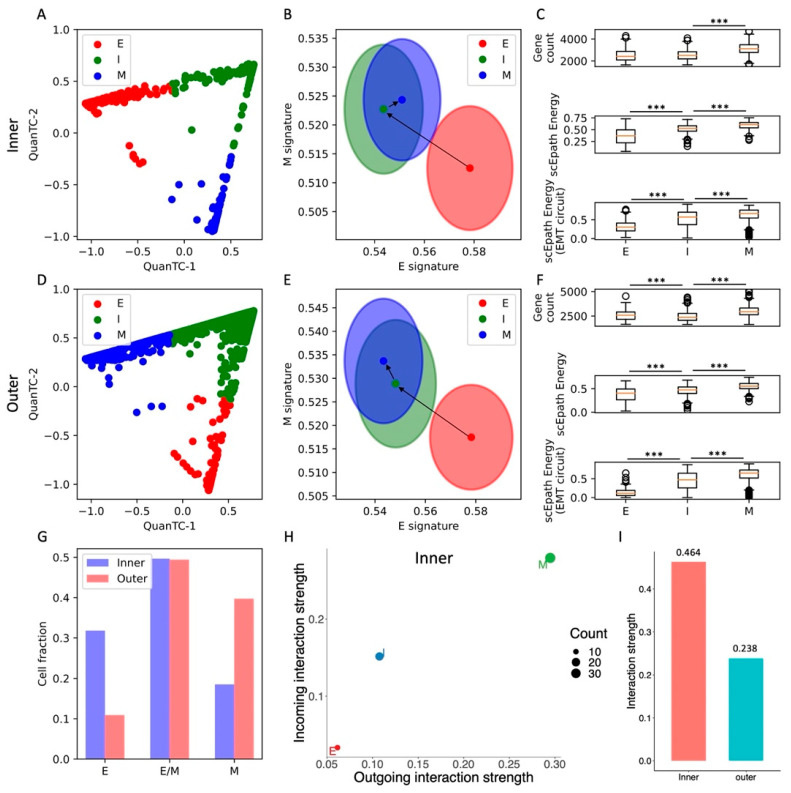
EMT characterization of inner and outer cell populations in a MCF10A 2D cell colony. (**A**) Low-dimensional representation of cells in the MCF10A-Inner dataset. *x*- and *y*-axis represent QuanTC coordinates. (**B**) Projection of clusters in the E-M signature space. The centering and radii of each circle represent the average scores and standard deviations of the epithelial and mesenchymal signatures within each cluster. Black arrows show the dominant transition trajectory predicted by QuanTC. (**C**) Boxplots depicting the transcriptional diversity (top), developmental potential (middle), and EMT circuit energy (bottom) in the four clusters. (**D**–**F**) Same as (**A**–**C**) for the MCF10A-Outer dataset. (**G**) Comparison of cell fractions belonging to the E, I, and M clusters in the inner and outer datasets. (**H**) Low-dimensional projection of cell clusters from the inner dataset in an incoming/outgoing signaling strength space computed with CellChat. (**I**) Comparison of total interaction strengths between the inner and outer datasets. This quantification considers all clusters and all signaling pathways. *** indicates *p*-value < 0.001 on a one-tailed *t*-test.

**Figure 5 cancers-13-05726-f005:**
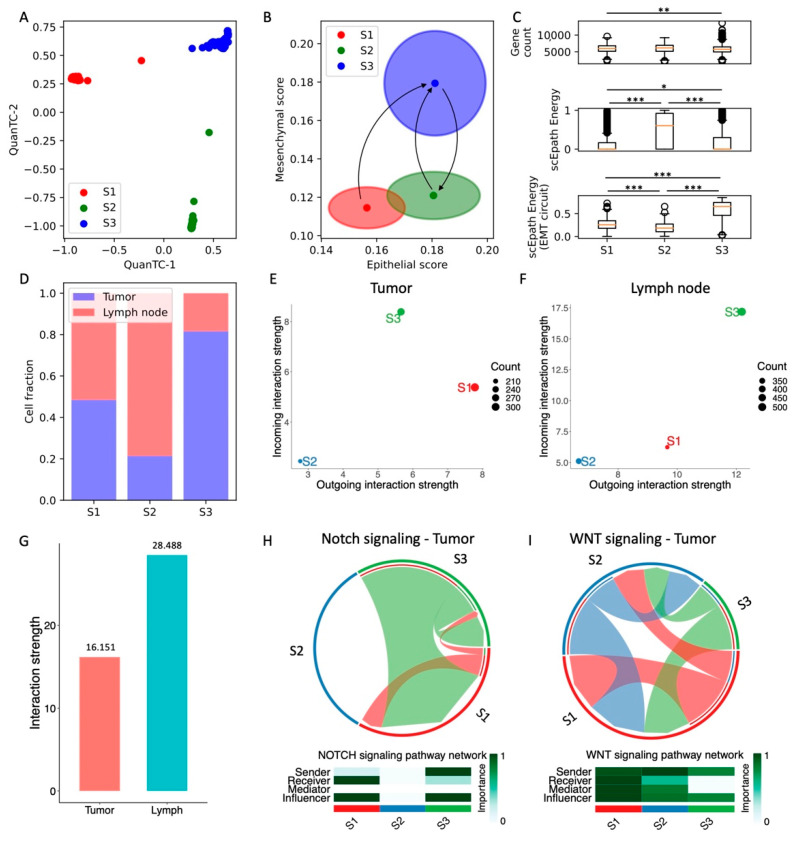
Analysis of head and neck SCC cells in tumor and lymph nodes. (**A**) Low-dimensional representation of cells in the HNSCC dataset. *x*- and *y*-axis represent QuanTC coordinates. (**B**) Projection of clusters in the E-M signature space. The center and radius of each circle represent the average scores and standard deviations of the epithelial and mesenchymal signatures within each cluster. Black arrows show the transition trajectories predicted by QuanTC when choosing different clusters as starting points. (**C**) Boxplots depicting the transcriptional diversity (top), developmental potential (middle), and EMT circuit energy (bottom) in the four clusters. (**D**) Fraction of cells sequenced from primary tumors (blue) and lymph node metastases (red) for the three clusters. (**E**,**F**) Low-dimensional projection of cell clusters from the tumor (**E**) and lymph node (**F**) datasets in an incoming/outgoing signaling strength space computed with CellChat. (**G**) Overall signaling strength in the tumor and lymph node cells quantified by CellChat. (**H**,**I**) Chord diagrams and signaling roles for the Notch (**H**) and WNT (**I**) pathways in tumor cells. *, **, and *** indicate a *p*-value < 0.1, 0.01, and 0.001 on a one-tailed *t*-test, respectively. For this dataset, all pairs of clusters were compared because no clear EMT trajectory was identified by QuanTC.

## Data Availability

Squamous cell carcinoma (SCC) dataset is available under accession number GSE110357. Time course of OVCA420 data under various inducers is available under accession number GSE147405. MCF10A data is available under accession number GSE114687. Head and neck squamous cell carcinoma (HNSCC) data is available under accession number GSE103322.

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
