# Peer review of "Single-Cell RNA-Seq Analysis Reveals the Acquisition of Cancer Stem Cell Traits and Increase of Cell–Cell Signaling during EMT Progression"

_cancers, 2021, doi:10.3390/cancers13225726_

Round 1

Reviewer 1 Report

This study shows reanalysis of cancer cell scRNA-seq.
Their results show the relationship between cancer-stem cell and EMT with single-cell resolution in several cancer cell models.
There are several concerns in the presented work.

Comments
1. The authors should show the significant gene expression of CSC, and cluster marker in each clusters using a heatmap for the all reanalysis of scRNA-seq. 

2. Probably, In Fig1F, G, H, Fig5H, I, Interal-sites of these circos plots need the edges which shown interaction between the clusters. In addition, Several presented figures have white blank(e.g. Fig 3B, Fig3E, Fig4G). Therefore, these are not able to confirm the results. The manuscript is not prepared well for the review. In most of figures and supplementary figures, I can not read the axis characters and sentences; therefore, I can not confirm the results. The resolution should be increased to see the figures.

3. The authors should show the basic information, the cell number of each cluster, results of clustering (PCA) method, and other significant marker genes(except the EM genes) in each clusters.

5. The manuscripts lacks the statistical viewpoints. The authors should perform the statistical test, and discuss the results.

6. The authors choice several scRNA-seq datasets from ovarian cancer cell lines, normal breast epithelial cell lines, HNSCC and others; however, The reasons to select are obscure and unclear. The authors should describe the reasons and choice the reasonable datasets. Their selection of datasets makes their study's design to be unclear. 

7. In relationship between EMT and CSC has already been reported by many previous studies(Wang et al., Clin Cancer Res., 2021; Simeorov et al., Cancer Cell, 2021; Deshmukh et al., PNAS, 2021;Nam et al., Nat Rev Genet., 2021 and more). The manuscript do not cite the these previous studies. The authors should cite the previous studies, and discuss it. In addition, Based on the understandings of previous researches, the authors should describe the novelty of presented work.

Reviewer 2 Report

This paper focuses on single cell rnaseq analysis of epithelial to mesenchymal transition to identify intermediate states and signaling pathways during EMT. Strengths of the study include the analysis of multiple datasets and the development of an EMT circuit energy gene expression analysis approach that is more reliable than traditional stem cell signatures for defining EMT. The circuit energy is a robust marker for EMT across in vitro and in vivo datasets .

Concerns:
  1. is the clustering robust across different clustering algorithm parameters? Please do an analysis where the clustering parameters are varied to show that clustering is consistent across different clustering algorithm parameters and not just an artifact of the algorithm parameters.
  2. There is no functional validation of any of the signaling findings from the cellchat analysis. At the very least it would be helpful to have flow cytometry data showing protein level expression of the ligands and/ or receptors in the specific stages of EMT
  3. The authors mention that the EMT did not completely reverse 3 days after EMT inducer withdrawal and that this may be due to hysteresis- it is not clear what this “hysteresis” means. Please explain this better.

Reviewer 3 Report

In this manuscript, Federico Bocci and colleagues address acquisition of cancer stem cell traits and EMT progression of cancer cells by using Single cell RNA-seq analysis. The authors used SCC, HNSCC, MCF10A, Ovarian cancer (OVCA420) cancer models to address CSC traits and EMT transition

However, the paper has a few significant weaknesses, outlined below:

  1. What does author mean to less frequent trajectories in the statement, would it be transition from I2 to M, or I1 to M, or E to M? And explain is transition possible directly from E to M?

                        Consistently, the dominant path involving the highest number of cells starts from the epithelial clusters, passes through two intermediate states, and finally reaches the state with the lowest epithelial signature (Fig. 1B). Other, less frequent trajectories are possible, which pass through only one or none of the intermediates before reaching the same terminal state. (206-to-210-page lines).

  1. Fig 1.B, Authors used AUcell to evaluate gene clusters i.e., E, I1, I2 and M. It would be helpful to reader what are all genes were analyzed in those clusters. Provide them in the supplementary data, which cluster has which set of genes.
  2. Fig 1.C, Upper fig: Average gene count decreased from E state to I1 state and increased, whether these four gene sets match to Fig 1.B or independent gene set. If it is different set of genes mentioned those gene sets.

Middle (ScPath Energy), Bottom (EMT circuit) Fig what those gene clusters are they same or different.

  1. Fig 1.D, Looks like transition from E to M is clear, what gene set has decreased in TN, whether it follows your observation Fig 1.B cluster gene sets? Mentioned this in the manuscript, it helps readers to understand about E to M transition.
  2. Whether JAG ligand is present in the all the cluster or it is expressed when the cells were acquired to I2 state?; TGF-Beta, WNT and Notch signaling is appeared when cells move from Epithelial state to I1, I2 and M. Whether any epigenetic markers are appeared in the transition or only transcriptional diversity observed?
  3. Fig 2.D Interestingly, the cell 297 population after 3 days from TGF-beta removal does not reverse completely to the original 298 cell state proportions but is rather a mixture of the four phenotypes (Fig. 2D)- Author’s statement (297-to-299-page line).

Can Authors explain this, If cells has any epigenetic profile changes, cells won’t reaches to previous state, so did Authors identified any epigenetic marker related gene expression in the scRNA-seq data in cell population after 3days of TGF beta removal.

  1. Fig 1 talk about the mouse SCC gene sets and Authors identified different clusters, where as Fig 2 is about human ovarian cancer OVCA420 cell line, and Fig 2A-B also has gene clusters. Both are cancer models, but different species. Here question when cancer transforming Epithelial to Mesenchymal state via Intermediate state whether both share same gene cluster or they are different.
  2. It would be easy for the readers if Authors provide gene sets related to those clusters, like E-cluster gene set, I1-cluster gene set, I2-cluster gene set and M-cluster gene set. Instead of Upregulated and downregulated gene set which authors provided in supplementary data and it wont talk about to which species gene sets they were. Since Authors worked on mouse (Fig 1) and Human (Fig 2)
  3. Fig 1, Cell-chat found JAG is the ligand for NOTCH signaling inducer identified (Need to know whether it is there every gene cluster or once cell start transition). Whether this ligand Identified Fig 2. Cluster? it would be better to comment about this in the results panel for fig2.
  4. Fig 3. It would be helpful to readers about cluster associated genes if authors mention them in manuscript. By changing the stimulant TNF (Fig 3D) and EGF (Fig 3A) Intermediate gene cluster didn’t change much, but Epithelial associated genes look very different, there is converging pathways to express Intermediate genes. It would be helpful for understand genes associated with this cluster.
  5. Fig 3B and Fig 3E-. Can’t interpret data Graph is missing.
  6. Fig 4. Defined genes associated with the cluster, if they were same with Fig1, Fig 2, Fig 3 cluster gene set, mention them in supplementary data.
  7. Fig 4G. Can’t interpret data Graph is missing.
  8. Fig 4, is about MCF10A cell line, which is non-tumorigenic cells, will they express Mesenchymal cluster genes? Without transforming them to tumorigenic cells.
  9. Fig 4, What is the status of WNT, NOTCH, and TGF signaling in MCF10A clusters in both outer and Inner region cells.
  10. Fig 5, E to M transition quite different in HNSCC as compared to other cancer models in this article, define S1, S2 and S3 gene sets.
  11. Fig 5D. Can’t interpret data Graph is missing.

Round 2

Reviewer 1 Report

The authors addressed all my concerns.

Author Response

We thank the reviewer for the useful comments.

Reviewer 2 Report

The manuscript has been made more clear and the authors have provided evidence that clustering results are robust across clustering parameters. However, there is zero validation fo the computational data. I agree that a full scale functional mechanistic validation of the computational data is beyond the scope of the manuscript (eg. signaling studies, culture studies, inhibitor studies). However, some form of minimal validation of the computational data (eg. flow cytometry for protein expression) is needed  to confirm that the computational results have real biological significance. In the absence of such validation it is impossible to know if the results have any biological meaning. 

Author Response

We are glad that the reviewer finds the manuscript clearer and is satisfied by our analysis of clustering robustness. We also appreciate the reviewer’s input concerning the experimental validation of the computational data. While we agree with these observations, there are scientific and technical limitations that prevent from fully addressing this comment. We have elaborated more on these limitations and how we addressed the comments below.

First, the main emerging result from the cell-cell signaling analysis is the correlative trend between EMT progression and the increased strength of cell-cell signaling. This trend does not emerge on a dataset-specific or pathway-specific basis, but rather when considering many pathways together. Secondly, on more concrete terms, we are a theoretical and computational lab without the capability to plan and perform the suggested experiments. In the ever-going feedback between experimental and theoretical investigation, we made use of some very interesting datasets and in turn provided new predictions and hypothesis that now somebody in the experimental community can take on and challenge.

We agree with the reviewer that analysis of scRNA-seq data in the context of cell-cell signaling has some intrinsic limitations, which were perhaps not fully addressed in the previous version of the manuscript. We have updated our final discussion to be even more explicit about these limitations: “Finally, cell-cell signaling takes place at the protein level; therefore, it is tacitly assumed that mRNA counts offer an accurate picture of their corresponding protein expression. Different technologies for direct protein level measurements, such as flow cytometry, will be required in the future to offer more biological context and demonstrate the validity of the predicted EMT-signaling association on a pathway-specific and/or tumor-specific basis”.

Moreover, these limitations might have been amplified by the fact that we have highlighted our findings on some of the pathways (Notch, WNT, TGFB) in effort to provide readers with “household” signaling pathways that are often discussed in cancer biology. However, we stress that our main result from the CellChat analysis is that the strength of cell-cell signaling increases along EMT, as a general trend,  independently of the specific pathway. This is also why we do not cite any specific pathway in the abstract and final conclusions. We have updated the discussion to make this clearer: “While several “usual suspects” often implicated in cancer progression, such as TGF-beta, WNT and Notch, were significantly expressed across multiple datasets, this association emerged as a general trend independently of the specific pathway”.

Round 3

Reviewer 2 Report

The utility of a study with no biological validation 

is very limited. The article in its current form is 

more suitable for a theoretical biology journal.